# Knowledge Extraction with No Observable Data

**Jaemin Yoo**
Seoul National University
jaeminyoo@snu.ac.kr

**Minyong Cho**
Seoul National University
chominyong@gmail.com

**Taebum Kim**
Seoul National University
k.taebum@snu.ac.kr

**U Kang**[*]
Seoul National University
ukang@snu.ac.kr

## Abstract

Knowledge distillation is to transfer the knowledge of a large neural network into a smaller one and has been shown to be effective especially when the amount of training data is limited or the size of the student model is very small. To transfer the knowledge, it is essential to observe the data that have been used to train the network since its knowledge is concentrated on a narrow manifold rather than the whole input space. However, the data are not accessible in many cases due to the privacy or confidentiality issues in medical, industrial, and military domains. To the best of our knowledge, there has been no approach that distills the knowledge of a neural network when no data are observable. In this work, we propose KEGNET (Knowledge Extraction with Generative Networks), a novel approach to extract the knowledge of a trained deep neural network and to generate artificial data points that replace the missing training data in knowledge distillation. Experiments show that KEGNET outperforms all baselines for data-free knowledge distillation. We provide the source code of our paper in https://github.com/snudatalab/KegNet.

## 1 Introduction

*How can we distill the knowledge of a deep neural network without any observable data?*

Knowledge distillation [9] is to transfer the knowledge of a large neural network or an ensemble of neural networks into a smaller network. Given a set of trained *teacher* models, one feeds training data to them and uses their predictions instead of the true labels to train the small *student* model. It has been effective especially when the amount of training data is limited or the size of the student model is very small [14, 28], because the teacher's knowledge helps the student to learn efficiently the hidden relationships between the target labels even with a small dataset.

However, it is essential for knowledge distillation that at least a few training examples are observable, since the knowledge of a deep neural network does not cover the whole input space; it is focused on a manifold $p_x$ of data that the network has actually observed. The network is likely to produce unpredictable outputs if given random inputs that are not described by $p_x$, misguiding the student network. There are recent works for distilling a network's knowledge by a small dataset [22] or only metadata at each layer [23], but no approach has successfully distilled the knowledge without any observable data. It is desirable in this case to generate artificial data by generative networks [7, 31], but they also require a large amount of training data to estimate the true manifold $p_x$.

We propose KEGNET (Knowledge Extraction with Generative Networks), a novel architecture that extracts the knowledge of a trained neural network for knowledge distillation without observable data.

---

[*]Corresponding author.

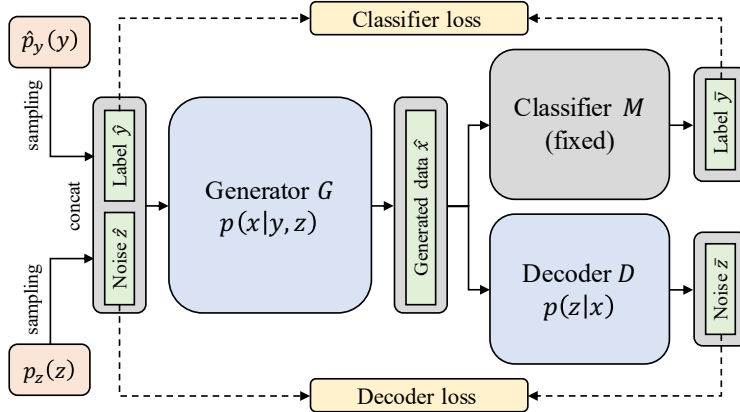

Figure 1: An overall structure of KEGNET. The generator creates artificial data and feed them into the classifier and decoder. The fixed classifier produces the label distribution of each data point, and the decoder finds its hidden representation as a low-dimensional vector.

KEGNET estimates the data manifold $p_x$ by generating artificial data points, based on the conditional distribution $p(y|x)$ of label $y$ which has been learned by the given neural network. The generated examples replace the missing training data when distilling the knowledge of the given network. As a result, the knowledge is transferred well to other networks even with no observable data.

The overall structure of KEGNET is depicted in Figure 1, which consists of two learnable networks $G$ and $D$. A trained network $M$ is given as a teacher, and we aim to distill its knowledge to a student network which is not included in this figure. The first learnable network is a generator $G$ that takes a pair of sampled variables $\hat{y}$ and $\hat{z}$ to create a fake data point $\hat{x}$. The second network is a decoder $D$ which aims to extract the variable $\hat{z}$ from $\hat{x}$, given as an input for $\hat{x}$. The variable $\hat{z}$ is interpreted as a low-dimensional representation of $\hat{x}$, which contains the implicit meaning of $\hat{x}$ independently from the label $\hat{y}$. The networks $G$ and $D$ are updated to minimize the reconstruction errors between the input and output variables: $\hat{y}$ and $\bar{y}$, and $\hat{z}$ and $\bar{z}$. After the training, $G$ is used to generate fake data which replace the missing training data in the distillation; $D$ is not used in this step.

Our extensive experiments in Section 5 show that KEGNET accurately extracts the knowledge of a trained deep neural network for various types of datasets. KEGNET outperforms baseline methods for distilling the knowledge without observable data, showing a large improvement of accuracy up to 39.6 percent points compared with the best competitors. Especially, KEGNET generates artificial data whose patterns are clearly recognizable by extracting the knowledge of well-known classifiers such as a residual network [8] trained with image datasets such as MNIST [21] and SVHN [25].

## 2 Related Work

### 2.1 Knowledge Distillation

Knowledge distillation [9] is a technique to transfer the knowledge of a large neural network or an ensemble of neural networks into a small one. Given a teacher network $M$ and a student network $S$, we feed training data to $M$ and use its predictions to train $S$ instead of the true labels. As a result, $S$ is trained by soft distributions rather than one-hot vectors, learning latent relationships between the labels that $M$ has already learned. Knowledge distillation has been used for reducing the size of a model or training a model with insufficient data [1, 2, 12, 14, 29].

Recent works focused on the distillation with insufficient data. Papernot et al. [28] and Kimura et al. [15] used knowledge distillation to effectively use unlabeled data for semi-supervised learning when a trained teacher network is given. Li et al. [22] added a $1 \times 1$ convolutional layer at the end of each block of a student network and aligned the teacher and student networks by updating those layers when only a few samples are given. Lopes et al. [23] considered the case where the data were not observable, but metadata were given for each activation layer of the teacher network. From the given metadata, they reconstructed the missing data and used them to train a student network.

However, these approaches assume that at least a few labeled examples or metadata are given so that it is able to estimate the distribution of missing data. They are not applicable to situations where no data are accessible due to strict privacy or confidentiality. To the best of our knowledge, there has been no approach that works well without training data in knowledge distillation, despite its large importance in various domains that impose strict limitations for distributing the data.

## 2.2 Tucker Decomposition

A tensor decomposition is to represent an $n$-dimensional tensor as a sequence of small tensors. *Tucker decomposition* [13, 30] is one of the most successful algorithms for a tensor decomposition, which decomposes an $n$-dimensional tensor $\mathcal{X} \in \mathbb{R}^{I_1 \times I_2 \times \cdots \times I_n}$ into the following form:

$$\hat{\mathcal{X}} = \mathcal{G} \times_1 \boldsymbol{A}^{(1)} \times_2 \boldsymbol{A}^{(2)} \times_3 \cdots \times_N \boldsymbol{A}^{(N)}, \tag{1}$$

where $\times_i$ is the $i$-mode product [16] between a tensor and a matrix, $\mathcal{G} \in \mathbb{R}^{R_1 \times R_2 \times \cdots \times R_n}$ is a core tensor, and $\boldsymbol{A}^{(i)} \in \mathbb{R}^{I_i \times R_i}$ is the $i$-th factor matrix.

Tucker decomposition has been used to compress various types of deep neural networks. Kim et al. [13] and Kholiavchenko [11] compressed convolutional neural networks using Tucker-2 decomposition which decomposes convolution kernels along the first two axes (the numbers of filters and input channels). They used the global analytic variational Bayesian matrix factorization (VBMF) [24] for selecting the rank $R$, which is important to the performance of compression. Kossaifi et al. [17] used Tucker decomposition to compress fully connected layers as well as convolutional layers.

Unlike most compression algorithms [3, 4], Tucker decomposition itself is a data-free algorithm that requires no training data in the execution. However, a fine-tuning of the compressed networks has been essential [11, 13] since the compression is done layerwise and the compressed layers are not aligned with respect to the target problem. In this work, we use Tucker decomposition to initialize a student network that requires the teacher's knowledge to improve its performance. Our work can be seen as using Tucker decomposition as a general compression algorithm when the target network is given but no data are observable, and can be extended to other compression algorithms.

## 3 Knowledge Extraction

We are given a trained network $M$ that predicts the label of a feature vector $x$ as a probability vector. However, we have no information about the data distribution $p_x(x)$ that was used to train $M$, which is essential to understand its functionality and to use its learned knowledge in further tasks. It is thus desirable to estimate $p_x(x)$ from $M$, which is the opposite of a traditional learning problem that aims to train $M$ based on observable $p_x$. We call this *knowledge extraction*.

However, it is impracticable to estimate $p_x(x)$ directly since the data space $\mathbb{R}^{|x|}$ is exponential with the dimensionality of data, while we have no single observation except the trained classifier $M$. We thus revert to sampling data points and modeling an empirical distribution: for a set $\mathcal{D}$ of sampled points, the probability of each sampled point is $1/|\mathcal{D}|$, and the probability at any other point is zero. We generate the set $\mathcal{D}$ of sampled data points by modeling a conditional probability of $x$ given two random vectors $y$ and $z$, where $y$ is a probability vector that represents a label, and $z$ is our proposed variable that represents the implicit meaning of a data point as a low-dimensional vector:

$$\mathcal{D} = \left\{ \arg\max_{\hat{x}} p(\hat{x}|\hat{y}, \hat{z}) \mid \hat{y} \sim \hat{p}_y(y) \text{ and } \hat{z} \sim p_z(z) \right\}, \tag{2}$$

where $\hat{p}_y(y)$ is an estimation of the true label distribution $p_y(y)$ which we cannot observe, and $p_z(z)$ is our proposed distribution that is assumed to describe the property of $z$.

In this way, we reformulate the problem as to estimate the conditional distribution $p(x|y, z)$ instead of the data distribution $p_x(x)$. Recall that $z$ is a low-dimensional representation of a data point $x$. The variables $y$ and $z$ are conditionally independent of each other given $x$, since they both depend on $x$ but have no direct interactions. Thus, the argmax in Equation (2) is rewritten as follows:

$$\arg\max_{\hat{x}} p(\hat{x}|\hat{y}, \hat{z}) = \arg\max_{\hat{x}} (\log p(\hat{y}|\hat{x}, \hat{z}) + \log p(\hat{x}|\hat{z}) - \log p(\hat{y}|\hat{z})) \tag{3}$$

$$= \arg\max_{\hat{x}} (\log p(\hat{y}|\hat{x}) + \log p(\hat{x}|\hat{z})), \tag{4}$$

where the first probability $p(\hat{y}|\hat{x})$ is the direct output of $M$ when $\hat{x}$ is given as an input, which we do not need to estimate since $M$ is already trained. The second probability $p(\hat{x}|\hat{z})$ represents how well $\hat{z}$ represents the property of $\hat{x}$ as its low-dimensional representation.

However, estimating the distribution $p(x|z)$ requires knowing $p_x(x)$ in advance, which we cannot observe due to the absence of accessible data. Thus, we rewrite Equation (4) as Equation (5) and then approximate it as Equation (6) ignoring the data probability $p_x(x)$:

$$\arg\max_{\hat{x}} p(\hat{x}|\hat{y},\hat{z}) = \arg\max_{\hat{x}}(\log p(\hat{y}|\hat{x}) + \log p(\hat{z}|\hat{x}) + \log p(\hat{x}) - \log p(\hat{z})) \tag{5}$$

$$\approx \arg\max_{\hat{x}}(\log p(\hat{y}|\hat{x}) + \log p(\hat{z}|\hat{x})). \tag{6}$$

The difference is that now we estimate the likelihood $p(\hat{z}|\hat{x})$ of the variable $\hat{z}$ given $\hat{x}$ instead of the posterior $p(\hat{x}|\hat{z})$. Equation (6) is our final target of estimation for extracting the knowledge of the given model $M$. We introduce in the next section how to model these conditional distributions by deep neural networks and how to design an objective function which we aim to minimize.

## 4 Proposed Method

KEGNET (Knowledge Extraction with Generative Networks) is our novel architecture to distill the knowledge of a neural network without using training data, by extracting its knowledge as a set of artificial data points of Equation (2). KEGNET uses two kinds of deep neural networks to model the conditional distributions in Equation (6). The first is a generator $G$ which takes $\hat{y}$ and $\hat{z}$ as inputs and returns a data point with the maximum conditional likelihood $p(\hat{x}|\hat{y},\hat{z})$. The second is a decoder $D$ which takes a data point $\hat{x}$ as an input and returns its low-dimensional representation $\bar{z}$.

The overall structure of KEGNET is depicted in Figure 1. The generator $G$ is our main component that estimates the empirical distribution by sampling a data point $\hat{x}$ several times. Given a sampled class vector $\hat{y}$ as an input, $G$ is trained to produce data points that $M$ is likely to classify as $\hat{y}$. This makes $G$ learn different properties of different classes based on $M$, but leads it to generate similar data points for each class. To address this problem, we train $G$ also to minimize the reconstruction error between $\hat{z}$ and $\bar{z}$, forcing $G$ to embed the information of $\hat{z}$ in the generated data $\hat{x}$ so that $D$ can successfully recover it. Thus, data points of the same class should be different from each other when given different input variables $\hat{z}$. The reconstruction errors are computed for $\hat{y}$ and $\hat{z}$, respectively, and then added to the final objective function. We also introduce a diversity loss to further increase the data diversity in each batch so that the generated data cover a larger region in the data space.

### 4.1 Objective Function

We formulate the conditional probabilities of Equation (6) as loss terms to train both the generator $G$ and decoder $D$, and combine them as a single objective function:

$$l(\mathcal{B}) = \sum_{(\hat{y},\hat{z})\in\mathcal{B}} \left( l_{\mathrm{cls}}(\hat{y},\hat{z}) + \alpha l_{\mathrm{dec}}(\hat{y},\hat{z}) \right) + \beta l_{\mathrm{div}}(\mathcal{B}), \tag{7}$$

which consists of three different loss functions $l_{\mathrm{cls}}$, $l_{\mathrm{dec}}$, and $l_{\mathrm{div}}$. $\mathcal{B}$ is a batch of sampled variables $\{(\hat{y},\hat{z}) \mid \hat{y} \sim \hat{p}_y(y), \hat{z} \sim p_z(z)\}$, and $\alpha$ and $\beta$ are two nonnegative hyperparameters that adjust the balance between the loss terms. Each batch is created by sampling $\hat{y}$ and $\hat{z}$ randomly several times from the distributions $\hat{p}_y$ and $p_z$ which are determined also as hyperparameters. In our experiments, we set $\hat{p}_y$ to the categorical distribution that produces one-hot vectors as $\hat{y}$, and $p_z$ to the multivariate Gaussian distribution that produces standard normal vectors.

The classifier loss $l_{\mathrm{cls}}$ in Equation (8) represents the distance between the input label $\hat{y}$ given to $G$ and the output $M(G(\hat{y},\hat{z}))$ returned from $M$ as the cross-entropy between two probability distributions. Note that $\hat{y}$ is not a scalar label but a probability vector of length $|\mathcal{S}|$ where $\mathcal{S}$ is the set of classes. Minimizing $l_{\mathrm{cls}}$ forces the generated data to follow a manifold that $M$ is able to classify well. The learned manifold may be different from $p_x$, but is suited for extracting the knowledge of $M$.

$$l_{\mathrm{cls}}(\hat{y},\hat{z}) = -\sum_{i\in\mathcal{S}} \hat{y}_i \log M(G(\hat{y},\hat{z}))_i \tag{8}$$

The decoder loss $l_{\mathrm{dec}}$ in Equation (9) represents the distance between the input variable $\hat{z}$ given to $G$ and the output $D(G(\hat{y},\hat{z}))$ returned from $D$. We use the Euclidean distance instead of the cross

entropy since $z$ is not a probability distribution. If we optimize $G$ only for $l_{cls}$, it is likely to produce similar data points for each class with little diversity. $l_{dec}$ prevents such a problem by forcing $G$ to include the information of $\hat{z}$ along with $\hat{y}$ in the generated data.

$$l_{dec}(\hat{y}, \hat{z}) = \|\hat{z} - D(G(\hat{y}, \hat{z}))\|_2^2. \tag{9}$$

However, the diversity of generated data points may still be insufficient even though $D$ forces $G$ to include the information of $\hat{z}$ in $\hat{x}$. In such a case, the empirical distribution estimated by $G$ covers only a small manifold in the large data space, extracting only partial knowledge of $M$. The diversity loss $l_{div}$ is introduced to address the problem and further increase the diversity of generated data. Given a distance function $d$ between two data points, the diversity loss $l_{div}$ is defined as follows:

$$l_{div}(\mathcal{B}) = \exp\left(-\sum_{(\hat{y}_1, \hat{z}_1) \in \mathcal{B}} \sum_{(\hat{y}_2, \hat{z}_2) \in \mathcal{B}} \|\hat{z}_1 - \hat{z}_2\|_2^2 \cdot d(G(\hat{y}_1, \hat{z}_1), G(\hat{y}_2, \hat{z}_2))\right). \tag{10}$$

It increases the pairwise distance between sampled data points in each batch $\mathcal{B}$, multiplied with the distance between $\hat{z}_1$ and $\hat{z}_2$. This gives more weights to the pairs of data points whose input variables are more distant by multiplying the distance of noise variables as a scalar weight. The exponential function makes $l_{div}$ produce a positive value as a loss to be minimized. We set $d$ to the Manhattan distance function $d(x_1, x_2) = \|x_1 - x_2\|_1$ in our experiments.

## 4.2 Relations to Other Structures

**Autoencoders** The overall structure of KEGNET can be understood as an autoencoder that tries to reconstruct two variables $y$ and $z$ at the same time. It is specifically an *overcomplete* autoencoder which learns a larger embedding vector than the target variables, since $x$ is a learned representation and $y$ and $z$ are target variables by this interpretation. It is typically difficult to train an overcomplete autoencoder because it can recover the target variable in the representation and make a zero reconstruction error. However in our case, the trained classifier $M$ prevents such a problem because it acts as a strong regularizer over the generated representations by classifying their labels based on its fixed knowledge. Thus, $G$ needs to be trained carefully so that the generated representations fit as correct inputs to $M$, while containing the information of both $y$ and $z$.

**Generative adversarial networks** KEGNET is similar to generated adversarial networks (GAN) [7] in that a generator creates fake data to estimate the true distribution, and the generated data are fed into another network to be evaluated. The structure of $G$ is also motivated by DCGAN [31] and ACGAN [26] for generating image datasets. However, the main difference from GAN-based models is that we have no observable data and thus we cannot train a *discriminator* which separates fake data from the real ones. We instead rely on the trained classifier $M$ and guide $G$ indirectly toward the true distribution. The decoder $D$ in KEGNET can be understood as an adversarial model that hinders $G$ from converging to a naive solution, but it is not a direct counterpart of $G$. Thus, KEGNET can be understood as a novel architecture designed for the case where no observable data are available.

## 4.3 Knowledge Distillation

To distill the knowledge of $M$, we use the trained generator $G$ to create artificial data and feed them into both the teacher $M$ and student $S$. We apply the following two ideas to make $S$ explore a large space and to maximize its generalization performance. First, we use a set $\mathcal{G}$ of multiple generators instead of a single network. Since each generator is initialized randomly, each of the generators learns a data manifold that is different from those of the others. The number of generators is not limited because they do not require observable training data. Second, we set $\hat{p}_y$ to the elementwise uniform distribution which generates unnormalized probability vectors: $\hat{y}_i \sim \mathcal{U}(0, 1)$ for each $i$. This gives an uncertain evidence to $G$ and forces it to generate data points which are not classified easily by $M$, making $M$ produce soft distributions in which its knowledge is embedded well.

As a result, our loss function $l_{dis}$ to train $S$ by knowledge distillation is given as follows:

$$l_{dis}(\hat{y}, \hat{z}) = \sum_{G \in \mathcal{G}} \text{CE}(M(G(\hat{y}, \hat{z})), S(G(\hat{y}, \hat{z}))), \tag{11}$$

where CE denotes the cross entropy. Previous works for knowledge distillation use a *temperature* [9] to increase the entropy of predictions from $M$ so that $S$ can learn hidden relationships between the classes more easily. We do not use the temperature since the predictions of $M$ are soft already due to our second idea of using the elementwise uniform distribution as $\hat{p}_y$.

Table 1: Detailed information of datasets.

| Dataset | Features | Labels | Training | Valid. | Test | Properties |
|---|---|---|---|---|---|---|
| Shuttle | 8 | 7 | 38,062 | 5,438 | 14,500 | Unstructured |
| PenDigits | 16 | 10 | 6,557 | 937 | 3,498 | Unstructured |
| Letter | 16 | 26 | 14,000 | 2,000 | 4,000 | Unstructured |
| MNIST | $1 \times 28 \times 28$ | 10 | 55,000 | 5,000 | 10,000 | Grayscale images |
| Fashion MNIST | $1 \times 28 \times 28$ | 10 | 55,000 | 5,000 | 10,000 | Grayscale images |
| SVHN | $3 \times 32 \times 32$ | 10 | 68,257 | 5,000 | 26,032 | RGB images |

Table 2: Classification accuracy of KEGNET and the baseline methods on the unstructured datasets. We report the compression ratios of student models along with the accuracy of Tucker.

| Model | Approach | Shuttle | Pendigits | Letter |
|---|---|---|---|---|
| MLP | Original | 99.83% | 96.56% | 95.63% |
| MLP | Tucker (T) | 75.49% ($8.17\times$) | 26.44% ($8.07\times$) | 31.40% ($4.13\times$) |
| MLP | T+Uniform | $93.83 \pm 0.13\%$ | $80.21 \pm 0.98\%$ | $62.50 \pm 0.90\%$ |
| MLP | T+Gaussian | $94.00 \pm 0.06\%$ | $78.22 \pm 1.74\%$ | $76.80 \pm 1.84\%$ |
| MLP | **T+KEGNET** | $\mathbf{94.21 \pm 0.03\%}$ | $\mathbf{82.62 \pm 1.05\%}$ | $\mathbf{77.73 \pm 0.33\%}$ |

## 5 Experiments

We evaluate KEGNET on two kinds of networks and datasets: multilayer perceptrons on unstructured datasets from the UCI Machine Learning Repository[2], and convolutional neural networks on MNIST [21], Fashion MNIST [33], and SVHN [25]. The datasets are summarized as Table 1.

We compare KEGNET with baseline approaches for distilling the knowledge of a neural network without using observable data. The simplest approach is to use Tucker decomposition alone, but the resulting student is not optimized for the target problem because its objective is only to minimize the reconstruction error. The second approach is to fine-tune the student after Tucker decomposition using artificial data derived from a sampling distribution. If the distribution is largely different from the true distribution, this approach may even decrease the performance of the student from the first approach. We use the Gaussian distribution $\mathcal{N}(0, 1)$ and uniform distribution $\mathcal{U}(-1, 1)$.

In each setting, we train five generators with different random seeds as $\mathcal{G}$ and combine the generated data from all generators. We also train five student networks and report the average and standard deviation of classification accuracy for quantitative evaluation. We initialize the compressed weights of student networks by running the singular value decomposition on the original weights and update them by Tucker decomposition to minimize the reconstruction errors [18]. We also use the hidden variable $\hat{z}$ of length 10 in all settings, which is much smaller than the data vectors. We use a decoder network of the same structure in all settings: a multilayer perceptron of $n$ hidden layers with the ELU activation [5] and batch normalization. $n$ is chosen by the data complexity: $n = 1$ in MNIST, $n = 2$ in the unstructured datasets, and $n = 3$ in Fashion MNIST and SVHN.

### 5.1 Unstructured Datasets

We use unstructured datasets in the UCI Machine Learning Repository, for which previous works [6, 27] have established reliable standards of performances. We select three datasets which have at least three classes and ten thousand instances. We divide each dataset into training, validation, and test sets with the 7:1:2 ratios if the explicit training and test sets are not given. Otherwise, we divide the given training data into new training and validation sets.

We use a multilayer perceptron (MLP) as a classifier $M$, which has been used in [27] and contains ten hidden layers with the ELU activation function and dropout [32] of probability $0.15$. We create student networks by applying Tucker decomposition to all dense layers: the target rank is 5 in Shuttle and 10 in the others. We use an MLP as a generator $G$ of two hidden layers with the ELU activation

Table 3: Classification accuracy of KEGNET and the baselines on the image datasets. We report the compression ratios of student models along with the accuracy of Tucker. We use three variants of students for each dataset with different compression ratios.

| Dataset | Model | Approach | Student 1 | Student 2 | Student 3 |
|---|---|---|---|---|---|
| MNIST | LeNet5 | Original | 98.90% | 98.90% | 98.90% |
| MNIST | LeNet5 | Tucker (T) | 85.18% (3.62×) | 67.35% (4.10×) | 50.01% (4.49×) |
| MNIST | LeNet5 | T+Uniform | 95.48 ± 0.11% | 88.27 ± 0.07% | 69.89 ± 0.28% |
| MNIST | LeNet5 | T+Gaussian | 95.45 ± 0.15% | 87.70 ± 0.12% | 71.76 ± 0.18% |
| MNIST | LeNet5 | **T+KEGNET** | **96.32 ± 0.05%** | **90.89 ± 0.11%** | **89.94 ± 0.08%** |
| SVHN | ResNet14 | Original | 93.23% | 93.23% | 93.23% |
| SVHN | ResNet14 | Tucker (T) | 19.31% (1.44×) | 11.02% (1.65×) | 11.07% (3.36×) |
| SVHN | ResNet14 | T+Uniform | 33.08 ± 1.47% | 63.08 ± 1.77% | 23.83 ± 1.86% |
| SVHN | ResNet14 | T+Gaussian | 26.58 ± 1.61% | 60.22 ± 4.17% | 21.49 ± 2.96% |
| SVHN | ResNet14 | **T+KEGNET** | **69.89 ± 1.24%** | **87.26 ± 0.46%** | **63.40 ± 1.80%** |
| Fashion | ResNet14 | Original | 92.50% | 92.50% | 92.50% |
| Fashion | ResNet14 | Tucker (T) | 65.09% (1.40×) | 75.80% (1.58×) | 46.55% (2.90×) |
| Fashion | ResNet14 | T+Uniform | < 65.09% | < 75.80% | < 46.55% |
| Fashion | ResNet14 | T+Gaussian | < 65.09% | < 75.80% | < 46.55% |
| Fashion | ResNet14 | **T+KEGNET** | **85.23 ± 1.36%** | **87.80 ± 0.31%** | **79.95 ± 1.36%** |

and batch normalization. We also apply the non-learnable batch normalization after the output layer to restrict the output space to the standard normal distribution: the parameters $\gamma$ and $\beta$ [10] are fixed as 0 and 1, respectively. This is natural since most neural networks take standardized inputs.

Table 2 compares the classification accuracy of student networks trained by KEGNET and the baseline approaches on the unstructured datasets. All three approaches show large improvements of accuracy over Tucker, which applies Tucker decomposition without fine-tuning. This implies that even simple distributions are helpful to improve the performance of student networks when no training data are observable. Nevertheless, KEGNET shows the highest accuracy in all datasets.

## 5.2 Image Datasets

We use two well-known classifiers on the image datasets: LeNet5 [20] for MNIST and ResNet with 14 layers (referred to as ResNet14) [8] for Fashion MNIST and SVHN. We initialize the student networks by compressing the weight tensors using Tucker-2 decomposition [13] with VBMF [24]; we compress only the convolutional layers except the dense layers as [13]. Since the classifiers are convolutional neural networks that are optimized for image datasets, we use a generator that is similar to that of ACGAN [26], which consists of two fully connected layers followed by three transposed convolutional layers with the batch normalization after each layer.

### 5.2.1 Quantitative Analysis

We evaluate KEGNET by training three different student networks for each classifier. For LeNet5, we compress the last convolutional layer in Student 1 and the last two convolutional layers in Student 2. We then increase the compression ratio of Student 2 by decreasing the projection rank in Student 3. For ResNet14, we compress the last residual block which consists of two convolutional layers. We compress each of the convolutional layers in Students 1 and 2 and the both layers in Student 3.

Table 3 shows the classification accuracy of student networks trained by KEGNET and the baseline approaches. In MNIST where the dataset and classifier are both simple, the Uniform and Gaussian baselines also achieve high accuracy which is up to 21.8%p higher than that of Tucker. However, their accuracy gain becomes much lower in SVHN, and the accuracy becomes even lower than that of Tucker in Fashion MNIST, meaning that fine-tuning after Tucker decomposition is not helpful at all. This shows that simple random distributions fail with complex datasets whose manifolds are far from trivial distributions. We do not report their exact accuracy in Fashion because they keep

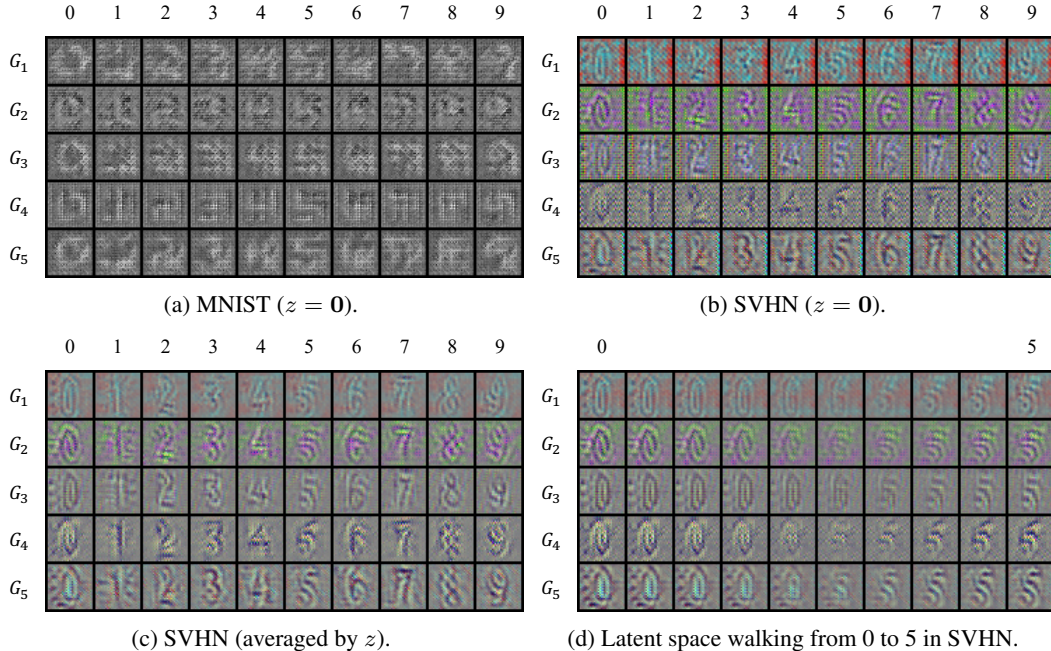

| | 0 | 1 | 2 | 3 | 4 | 5 | 6 | 7 | 8 | 9 |
| | 0 | 1 | 2 | 3 | 4 | 5 | 6 | 7 | 8 | 9 |

(a) MNIST ($z = \mathbf{0}$).　　　　　　(b) SVHN ($z = \mathbf{0}$).

(c) SVHN (averaged by $z$).　　　　　(d) Latent space walking from 0 to 5 in SVHN.

Figure 2: Artificial images generated by five generators of KEGNET for MNIST and SVHN. We fix the noise variable $z$ to a zero vector in (a) and (b), while we average multiple images with random $z$ in (c) and (d). The digits are blurry but recognizable especially when averaged by $z$.

decreasing as we continue the training. On the other hand, KEGNET outperforms all baselines by learning successfully the data distributions from the given classifiers in all datasets.

### 5.2.2 Qualitative Analysis

We also analyze qualitatively the extracted data for the image datasets. Figure 2 visualizes artificial images for MNIST and SVHN, which were generated by the five generators in $\mathcal{G}$. The images seem noisy but contain digits which are clearly recognizable, even though the generators do not have any information about the true datasets. KEGNET generates more clear images in SVHN than in MNIST, because the digits in SVHN have more distinct patterns than in the hand-written digits. The digits are more clear when averaged from multiple hidden variables, implying that images with different hidden variables are diverse but share a common feature that the classifier is able to capture. We also visualize images with soft evidence in Figure 2d by changing smoothly the input label from 0 to 5. It is shown that the generators create digits following the strength of evidence for each class.

## 6 Conclusion

We propose KEGNET (Knowledge Extraction with Generative Networks), a novel architecture that extracts the knowledge of a trained neural network without any observable data. KEGNET learns the conditional distribution of data points by training the generator and decoder networks, and estimates the manifold of missing data as a set of artificial data points. Our experiments show that KEGNET is able to reconstruct unobservable data that were used to train a deep neural network, especially for image datasets that have distinct and complex manifolds, and improves the performance of data-free knowledge distillation. Future works include extending KEGNET to knowledge distillation between neural networks of different structures, such as LeNet5 and ResNet14, or to more complex datasets such as CIFAR-10/100 [19] that may require a careful design of new generator networks.

**Acknowledgments**

This work was supported by the ICT R&D program of MSIT/IITP (No.2017-0-01772, Development of QA systems for Video Story Understanding to pass the Video Turing Test).

## Footnotes

[2]http://persoal.citius.usc.es/manuel.fernandez.delgado/papers/jmlr/

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
