[Reviews · NeurIPS 2019]

Reviewer 1



Originality: The proposed approach is a novel application and combination of GAN-style generative models, knowledge distillation and model Compression. Quality: The proposed approach seems like a technically sound and sensible approach to solving the stated problem. I’m a little confused by in section 4.3 where during knowledge distillation the target class is soft-sampled from a uniform distribution and fed un-normalized into the generator, as especially considering that it was not trained in this way. Perhaps a more consistent approach would be to do the same during both training and inference. Alternatively, you could sample categorical distributions from a carefully specified Dirichlet Prior and minimise the KL-divergence between the sampled categorical distribution and the categorical predicted by M instead in equation 8. This would be a less adhoc and more consistent approach to using soft-targets during training and inference. I’m concerned that since KegNet trained networks are initialised using SVD instead of Tucker-2, that you do not provide baseline numbers for networks compressed using SVD vs. Tucker-2. I would also be good to see use of KegNet to train a student networks which is initialised randomly without compression. Clarity: This submission is clearly written and informative. I could reproduce the results based on this paper (even without provided code). However, notation is a little overloaded. Would be good to specify exactly this regards to what parameters the losses in equations 7-10 are (this can still be inferred, but good to be explicit). It not entirely clear why you initialised KegNet-trained student Networks Minor points: It is not entirely clear in which parts of the student network are NOT compressed and simply carried over from the teacher. Also, In line 255 you refer to Resnet20, while in Table 3 you refer to Resnet14. Can you please clarify this discrepancy. Significance: This paper seems to provide a sensible solution to a novel problem. However, I have some concerns over the experimental results. The most complex model/dataset which was ResNet14 on FashionMNIST and SVHN. It would be good to see some results on CIFAR10/CIFAR-100, as MNIST/SVHN/FashionMNIST are still very simple datasets. Also, as described in the Quality section, I’m concerned about the lack of KetNet+ random init, SVD and SVD+finetuning numbers. ---POST-REBUTTAL COMMENTS--- The claimed bad performance on CIFAR-10 and CIFAR-100 undermines the significance of the method. However, I am happy to vote to accept this as a stepping stone to a more advanced methods, as long as the authors are very honest and explicit about the limitations of this approach. Regarding sampling unnormalised vectors - I am happy that this approach does improve performance, and the ablation study is quite useful. However, the mismatch between training and test time could be hampering the method, and eliminating it could yield further improvement in performances. Furthermore, I would be happier on a conceptual level with an approach which is a little more principled - I find the lack of normalisation confusing and unnecessary. The large standard deviation of idea 2 seems to support this. At the same time, I had no issues with an ensemble of generators - I though that was very sensible. Finally, it would be ideal if the authors could include the detailed description of how their method integrates with SVD and T2 into the main body of the paper as well as point out which parts of the network and retained and which are compressed. This added clarity would strengthen the paper.

Reviewer 2



The problem presented in the paper is very interesting and challenging, as it considers the case where no observable data is present, and the approach proposed is in some sense very reasonable, generating artificial data points. The theoretical explanation for this approach follows a dependence assumption and simple approximations and de-compositions of the probability p_x. The weakest part in this paper is the experimental part. Although the framework is new and lacks of fair benchmarks it seems that several details and experiments are lacking in order to understand the benefits of using this approach and its limitations: 1. Why the size of z was chosen to be 10? How does it affect the results? 2. Why z is assumed to be low dimensional? 3. How your method scales to more complex and challenging datasets?

Reviewer 3



novelty: paper addresses new problem with a new approach. I think the problem is interesting. Originally I was not convinced by the approach, given that this could lead to learning adversarial examples as well, but results seem to confirm the the validity of the proposed method (though I still have some doubts in point 5) quality: the proposed method is in general clear and should be easy to implement given the paper. Experimental results confirm the proposed approach. clarity: the paper was in general easy to follow, significance: I am unaware that this problem has been addressed before. Extracting knowledge from a network without access to data is potentially impactful.

[Author Response · NeurIPS 2019]

Table 1: Ablation study of proposed ideas for knowledge distillation.

| Ablation | Accuracy |
|---|---|
| Baseline | 11.07% |
| No ideas | $27.41 \pm 4.76\%$ |
| Idea 1 | $55.10 \pm 2.49\%$ |
| Idea 2 | $44.21 \pm 14.0\%$ |
| **Idea 1 & 2** | $\mathbf{63.40 \pm 1.80\%}$ |

Table 2: Classification accuracy by the training epochs of generators.

| Epochs | Accuracy |
|---|---|
| 0 | $23.21 \pm 1.25\%$ |
| 10 | $42.52 \pm 2.19\%$ |
| 50 | $52.03 \pm 2.51\%$ |
| 100 | $61.70 \pm 3.94\%$ |
| **200 (ours)** | $\mathbf{63.40 \pm 1.80\%}$ |

Table 3: Comparison between different lengths of noise variables $z$.

| Length | Accuracy |
|---|---|
| 8 | $49.23 \pm 3.02\%$ |
| **10 (ours)** | $\mathbf{63.40 \pm 1.80\%}$ |
| 12 | $60.65 \pm 1.29\%$ |
| 16 | $61.39 \pm 2.56\%$ |
| 20 | $59.29 \pm 0.84\%$ |

Thank you for the detailed reviews. We address your comments and attach additional experimental results. We group
the issues based on the topics and show the related reviewers for clarification: for instance, R1 denotes Reviewer 1. All
experiments in this letter have been done for Student 3 on the SVHN dataset.

**Tucker decomposition for initialization (R1, R3).** We initialize the student networks by Tucker-2 decomposition.
Specifically, we take three steps to train each student network: we 1) initialize the weights by running SVD on the
teacher networks, 2) update them by Tucker-2 to minimize the reconstruction errors, and 3) fine-tune them by artificial
data points from the generators. Since applying step 1 alone produces low accuracy, we take it as a baseline to apply
steps 1 and 2 together and report it in the paper as *Tucker (T)*. If we apply step 3 alone without steps 1 and 2, as Reviewer
1 suggested as *KegNet + random init*, the accuracy is not as good as shown in the paper since the student networks have
no prior knowledge about the learned weights of the teachers. It is a challenging problem to apply KegNet without
initializing the student networks by Tucker-2, as it is more difficult to train them properly.

**Soft labels by unnormalized distributions (R1, R3).** We propose two ideas in line 189 to improve the performance
of knowledge distillation. The first is to use multiple generators when generating artificial data points. The second
is to sample label vectors from the elementwise uniform distribution; instead of using a typical one-hot vector or a
categorical distribution as a label vector $\hat{y}$, we sample each element independently from $\mathrm{uniform}(0, 1)$ and create an
unnormalized probability vector as an input label. As a result, we do not impose any correlation between the different
classes but generate diverse data points that cover a larger manifold. Table 1 shows the result of ablation study of these
ideas. Currently, even a simple idea is enough to improve the performance of our model by a large margin, but we may
apply a more principled approach to achieve a similar objective following the suggestion of Reviewer 1.

**Performance improvements during training (R3).** Reviewer 3 commented that the superiority of our approach may
have come from the structural prior imposed by CNN-based generators. To address the concern, we report the accuracy
of student networks, coupled with various generators trained for different numbers of epochs. Table 2 clearly shows that
it is essential to train enough the generators to get a superior performance; this is because the artificial data generated
from random generators are not close enough to the true data manifold which we aim to estimate.

**The length of noise variables (R2).** The noise variable $z$ has been designed to follow the class-independent manifold
of the data distribution $p_x$; compare it with $y$ which embeds class-dependent manifold of the distribution. Thus, it is
reasonable to assume that $z$ lies in a low-dimensional embedding space as done in previous works [25]. At the same
time, it is important to choose a proper length of $z$ as it determines the capacity and learning complexity of our model.
Table 3 compares the accuracy of student networks trained with noise variables of different lengths. Accuracy is the
best when the length is 10 as in the paper, but the differences between different lengths are negligible compared with
the other experiments in Tables 1 and 2; this implies that our approach is not very sensitive to the length of $z$.

**More complex datasets (R1, R2).** We ran additional experiments for other datasets such as CIFAR10 and CIFAR100
which are larger and more complex than the datasets that we used in the paper. As a result, we have checked that it is
challenging to achieve a good performance on these datasets, because they have more complex data manifolds which
are difficult to be estimated by our simple KegNet structure. It seems that a more complex architecture is needed to
capture such a complex manifold, and thus we leave it as an open problem for future works.

**Minor points (R1, R2, R3).** (R1) We have typing errors in lines 245 and 246; we used ResNet14 instead of ResNet20.
(R1) We compressed only the convolutional layers in the teacher networks as described in line 252 and did not touch
the dense layers, based on previous works on compressing CNNs by Tucker-2 [13]. (R2) Our objective is to distill the
knowledge of a neural network in the absence of training data. This is done by generating artificial data that follow a
similar manifold to the unseen data, and the result can be used for various applications such as model compression,
interpretation, or knowledge transfer especially when the data are not accessible. (R3) We had visualized the generated
images for Fashion MNIST, but it was difficult to recognize the images because they described ambiguous clothes, hats
or shoes rather than clear digits. It is a future work to make the model generate more recognizable images.

[Meta-Review · NeurIPS 2019]

This paper proposes to solve the novel task of knowledge distillation in the absence of training data. Reviewer 1 thought the paper was a novel application and combination of GANs, knowledge distallation and model compression. They thought the approach was technically sound and sensible. They had some issues with the way to sample the target class and suggested a more consistent approach. They also highlighted some concerns with the absence of baselines for SVD-compressed networks with and without fine-tuning. They felt that overall the work was significant but expressed some concerns with the scale of experiments, noting that the most complex model/dataset was ResNet14 on FashionMNIST and SVHN. Reviewer 2 thought that the problem presented was interested in challenging and, like Reviewer 1, felt that there were some weaknesses in the experiments. Reviewer 2 also wanted to see more motivation to the problem presented. Reviewer 3 thought the problem was novel and relevant while significance was moderate due to the presence of related architectures. Both Reviewer 1 and 3 commented that the paper was clear and would be easy to re-implement. The authors responded to Reviewer 1 and 3’s request for clarity around the use of Tucker-2 decomposition for initialization. They also presented a further analysis of sampling label vectors. They also ran additional experiments on CIFAR10 and CIFAR100, reporting that is challenging to achieve a good performance there, leaving it to be an open problem. The reviewers felt that the author response improved clarity but still raised some questions, particularly with respect to the gains on unstructured data (emphasized by the poor performance on CIFAR). The reviewers and AC see no issue with the negative results being reported and agree on acceptance, with the paper a stepping stone to more advanced methods. We all recommend that the paper moderates its claims.